# Estrogen Signaling Induces Mitochondrial Dysfunction-Associated Autophagy and Senescence in Breast Cancer Cells

**DOI:** 10.3390/biology9040068

**Published:** 2020-04-01

**Authors:** Khuloud Bajbouj, Jasmin Shafarin, Jalal Taneera, Mawieh Hamad

**Affiliations:** 1Research Institute for Medical and Health Sciences, University of Sharjah, Sharjah 27272, UAE; kbajbouj@sharjah.ac.ae (K.B.); jsalam@sharjah.ac.ae (J.S.); jtaneera@sharjah.ac.ae (J.T.); 2Department of Basic Medical Sciences, College of Medicine, University of Sharjah, Sharjah 27272, UAE; 3Department of Medical Laboratory Sciences, College of Health Sciences, University of Sharjah, Sharjah 27272, UAE

**Keywords:** Estrogen, senescence, MCF-7, MDA-MB-231, mitochondria, autophagy

## Abstract

Previous work has shown that although estrogen (E2) disrupts cellular iron metabolism and induces oxidative stress in breast and ovarian cancer cells, it fails to induce apoptosis. However, E2 treatment was reported to enhance the apoptotic effects of doxorubicin in cancer cells. This suggests that E2 can precipitate anti-growth effects that render cancer cells more susceptible to chemotherapy. To investigate such anti-growth non-apoptotic, effects of E2 in cancer cells, MDA-MB-231 and MCF-7 cells were evaluated for the expression of key autophagy and senescence markers and for mitochondrial damage following E2 treatment. Treated cells experienced mitochondrial membrane depolarization along with increased expression of LC3-I/II, Pink1 and LAMP2, increased LC3-II accumulation and increased lysosomal and mitochondrial accumulation and flattening. E2-treated MCF-7 cells also showed reduced P53 and pRb780 expression and increased Rb and P21 expression. Increased expression of the autophagy markers ATG3 and Beclin1 along with increased levels of β-galactosidase activity and IL-6 production were evident in E2-treated MCF-7 cells. These findings suggest that E2 precipitates a form of mitochondrial damage that leads to cell senescence and autophagy in breast cancer cells.

## 1. Introduction

Numerous studies have shown that estrogen (17β-estradiol; E2) induces tumorigenesis in mammals and increases the risk of breast and ovarian cancer in humans [1,2,3,4]. That said, mounting evidence suggest that E2 signaling also precipitates anti-cancer effects in E2-responsive cancers of the breast, ovaries, uterus and liver. High-dose E2 treatment was successfully used against metastatic breast cancer in postmenopausal women for decades before the introduction of selective estrogen receptor modulators (SERMs) in the 1970s [5]. Clinical trials that evaluated the efficacy of tamoxifen versus high dose diethylstilbestrol (DES) against metastatic breast cancer have shown comparable clinical outcomes [6] with patients receiving DES surviving longer than those receiving tamoxifen [7]. E2 replacement therapy was also reported to reduce the incidence of invasive breast cancer in postmenopausal women with prior hysterectomy [8]. Furthermore, differences in the incidence of hepatocellular carcinoma, being 2–4 times higher in males than in females [9,10,11], has been attributed to the protective effects of E2 [12].

The anti-carcinogenic effects of E2 have been attributed mainly to its ability to induce apoptosis via E2/ER interactions with estrogen response elements (EREs) in genes of the intrinsic apoptotic pathway or via CD95-CD95L interactions that engage and activate the extrinsic apoptotic pathway [13]. Several studies have demonstrated that E2-induced apoptosis occurs in cancer cells that have been previously sensitized by E2 deprivation [13,14,15]. However, E2 treatment in unsensitized breast and ovarian cancer cells has been recently shown to disrupt the cell cycle and precipitate cellular damage without leading to apoptosis [15,16]. E2 signaling has also been recently shown to enhance the cytotoxic effects of chemotherapy (doxorubicin) in unsensitized breast and ovarian cancer cells [17]. These observations suggest that the anti-cancer effects of E2, which render cancer cells more susceptible to chemotherapy, may manifest in the form of autophagy and/or senescence. In this context, numerous studies over the last 10 years have elaborated on the intricate relationship between E2 signaling and autophagy in endocrine-related cancers [18]. Several studies have also reported on the involvement of senescence in tumorigenesis [19,20] and therapy [21] and on the differential effects of E2 on mitochondrial function [22,23,24,25]; a key driver of autophagy [18] and senescence [19,20]. To address the possibility that E2 induces senescence and autophagy in breast cancer cells, metastatic (MDA-MB-231) and non-metastatic (MCF-7) human breast cancer cells were grown in serum-supplemented media not subjected to charcoal stripping for over 20 generations to rule out E2 deprivation. Cells were then treated with 20 nM E2 and assessed for the expression of key markers of autophagy and cell senescence and for mitochondrial damage at different time points post-treatment. The choice to treat cells with 20 nM E2 is based on previous work which addressed the anti-cancer effects of E2 and its ability to induce oxidative stress, cellular damage and cell cycle arrest [15,16,17].

## 2. Materials and Methods

Cell lines and culture conditions: Human breast cancer cell lines MCF-7 (ER^+^) and MDA-MB-231 (ER^-^) (American Type Culture Collection (Manassas, VA, USA) were used throughout the study. Both MCF-7 and MDA-MB-231 cells were maintained in Dulbecco’s Modified Eagle’s Medium (DMEM) supplemented with 2 μg/mL of insulin, 1 mM of sodium pyruvate, 1 mM of nonessential amino acids, 4 mM of glutamine, 10% fetal calf serum (non-charcoal-stripped) and antibiotics (penicillin/streptomycin) at 37 °C and 5% CO_2_ for over 20 generations. Freshly-split cells were seeded at 0.5–1 × 10^5^ cells/mL in 25 cm flasks; at ~70% confluency and treated with 17-β estradiol (estradiol benzoate; Sigma-Aldrich, St. Louis, Missouri, United States) at 20 nM dissolved in ethanol; control cells were treated ethanol as vehicle. Where appropriate, cells were treated with bafilomycin (Cat No. B1793; Sigma-Aldrich) at 20 nM.

Mitochondrial proteins: Isolation of mitochondria from E2-treated and control cells was performed using a mitochondria isolation kit (Cat No. 89874; Thermo-Scientific, Pierce, USA). Briefly, cells were centrifuged at 850 XG for 2 min, cell suspension was harvested, 800 µL of mitochondria isolation reagent A was added and mixed by vortexing at medium speed for 5 s and incubated on ice for 2 min. 10 µL of mitochondria isolation reagent B was added and mixed by vortexing at maximum speed for 5 s; mixture was then incubated on ice for 5 min and vortexed at maximum speed every minute. Eight hundred microliters of mitochondria isolation reagent C was added, cells were then centrifuged at 700 XG for 10 min at 4 °C and supernatant was re-centrifuged at 3000 XG for 15 min at 4 °C. Supernatant (cytosolic fraction) was transferred to a new tube and stored at −80 °C until further use. Five hundred microliters mitochondria isolation reagent C was added to the pellet, centrifuged at 12,000 XG for 5 min, then the supernatant was discarded. The pellet was lysed with RIPA lysis buffer and lysate was stored at −80 °C prior to Western blotting.

Western blotting: Cells were lysed in ice-cold NP-40 lysis buffer (1.0% NP-40, 150 mM of NaCl, 50 mM of Tris-Cl, pH 8.0) containing protease cocktail inhibitor tablets (Cat. No. S8830; Sigma-Aldrich). Whole cell lysate protein concentrations were quantified by the Bradford method (Cat. No. 500-0006; Bio-Rad, Hercules, CA USA). Lysate aliquots containing 30 μg of protein were run on 12% sodium dodecyl sulfate–polyacrylamide gel electrophoresis (SDS-PAGE), transferred onto a nitrocellulose membrane (Cat. No. 1620112; Bio-Rad). Membrane was blocked with 5% skimmed milk powder for 1 h at room temperature, washed with (TBST), and reacted with primary unlabeled antibody (IgG) at 1:1000 dilution overnight at 4 °C. Primary antibody list included: anti-P53 (Cat. No. 2524), anti-P21 (Cat.No. 2947), anti-LC3-I/II (No. 12741), anti-Rb (Cat. No. 9969), anti p-Rb780 (No. 8180), PINK 1 (Cat. No. 6946) and Beclin 1 (Cat. No. 3495) and the autophagy antibody sampler kit (Cat. No. 4445), which included antibodies against ATG5, ATG12, ATG 16L1, ATG7, and ATG3 (Cell Signaling, Danvers, MA, USA). Antibodies against was purchased for LAMP-2 (Cat. No. AB125068; Abcam, Cambridge, UK). Secondary anti-mouse (Cat. No. 7076; Cell Signaling Technology, Anvers, MA, USA) was reacted with the membrane at 1:1000 dilutions for 1 h at room temperature and the secondary anti-rabbit antibody (Cat. No. 97040; Abcam) was reacted with the membrane at 1:5000 dilution for 1 h at room temperature. Chemiluminescence was detected using the ECL kit (Cat. No. 32106; Thermo-Scientific). Protein band quantification was carried out using the Bio-Rad Image Lab software (ChemiDoc™ Touch Gel and Western Blot Imaging System; Bio-Rad, Hercules, CA, USA). β-actin was used as a loading and normalization control. LC3-I-to-/II conversion rate was calculated as densitometry reading of LC3-II normalized against β-actin / densitometry reading of LC3-I normalized against β-actin.

Flow cytometry analysis: Cells (10^6^) were washed twice and then either directly stained (for cell surface markers detection) or fixed using Cytofix/Cytoperm Fixation/Permeabilization Solution Kit (BD, United States) (for intracellular markers detection). Cells were incubated for 20 min with stain buffer (Becton Dickinson, Franklin Lakes, NJ, USA) containing conjugated antibodies against; TNF-α or IL-6 (BioLegends, San Diego, CA, USA). Cells were then washed and analyzed using a BD FACS-Aria™ III flow cytometer (Becton-Dickinson, San Jose, CA, USA) at a rate of 1000 events/sec. A minimum of 25,000 events were collected/sample and percentage positive staining was computed to the 99% level of confidence. Flow cytometry data were analyzed using the *FlowJo* software with the Watson pragmatic model (Flowjo, Ashland, Oregon, USA). MFI represents the geometric mean of fluorescence signals.

Measurement of mitochondrial membrane *potential (MMP;* ΔΨm): a JC-1 Mitochondrial Membrane Potential Assay flow cytometry-based Kit (Abcam) was used according to manufacturer’s protocol. For quantification of JC-1 intensity, cells were seeded in a 96-well black plate with clear bottom. Ex 488/Em 530 nm and Ex 550/Em 600 nm were used and MMP was calculated as the ratio of red-to-green fluorescence.

Assessment of mitochondrial and lysosomal accumulation: Cells were seeded at a density of 5 × 10^5^ cells/mL; at around 60% confluency, cells were treated with 20 nM E2 for 24 and 48 h or left untreated. Cells were then harvested, washed twice with PBS and stained for nuclear DNA with DAPI along with the Mitotracker or Lysotracker stains according to manufacturer’s instructions (Invitrogen, Carlsbad, CA, USA). Cells were then stained with anti-LC3-I/II (Cat. No. 12741; Cell Signaling) at 1:1000 dilution overnight at 4 °C. Cells were then washed with 1X PBS and reacted with the Alexafluor^®^_680_-labeled secondary antibody (Abcam) for 1 h at 37 °C; excess reagent was rinsed with 1X PBS. Genomic DNA was stained with 4′,6′-diamidino-2-phenylindole (DAPI) (Cat. No. D1306, Invitrogen) according to manufacturer’s instructions. Slides were visualized by confocal microscopy using a Nikon Confocal Microscope (Nikon, Tokyo, Japan).

β-Galactosidase cell senescence assay: Accumulation of β-Galactosidase in E2-treated and control cells was assed using the colorimetric senescence-associated SA-β-Galactosidase (SA-β-Gal) assay kit according to manufacturer’s instructions (Cell Signaling).

Statistical analysis: Data sets representing protein quantitation, MMP, LIP, and cell count were analyzed using the online GraphPad Software (https://www.graphpad.com/quickcalcs/ttest2/). Unpaired student *t* test was used to generate *p* values for comparisons between groups in each data set; *p < 0.05* was considered significant.

## 3. Results

E2 treatment induces mitochondrial accumulation and autophagy in breast cancer cells: Several previous studies have reported on the ability of E2 to disrupt intracellular iron metabolism and to induce oxidative stress in breast cancer cells [15,16]. Previous work has also shown that this associates with cell cycle arrest and plasma membrane damage but not apoptosis [16,26]. In this study, we assessed mitochondrial functional integrity, accumulation and flattening along with the expression of key cell senescence and autophagy-related proteins in E2-treated cells as means of further understanding the anti-cancer non-apoptotic, effects of E2 in cancer. As shown in Figure 1A, MCF-7 cells treated with 20 nM E2 showed increased mitochondrial accumulation and increased expression of the auto-phagosome marker LC3-I/II, especially at 24 h post-treatment. Moreover, senescence-associated heterochromatin foci (SAHFs; indicated by white arrows in DAPI-stained cells) were evident in E2-tretaed MCF-7 cells at 24 and 48 h post-treatment. E2-treated MCF-7 cells also showed reduced expression of P53 and increased expression of P21 and LC3; this was particularly evident in cells treated with 20 nM E2 for 48 h (Figure 1B,C). Similar, though less pronounced, findings were observed in E2-treated MDA-MB-231 cells (Figure 2A–C). In that, the expression of P53, P21 and LC3-I/II increased in MDA-MB-231 cells at 24 h but decreased at 48 h post-treatment.

Expression pattern of autophagy- and cell cycle-related proteins in MCF-7 cells: To further investigate the autophagic effects of E2, the expression of several autophagy-related proteins was investigated in MCF-7 and MDA-MB-231 cells treated with 20 nM E2 for 24 and 48 h. As shown in Figure 3A, E2-treated MCF-7 cells showed increased expression of ATG3 and beclin, especially at 48 h post treatment. E2-treated MDA-MB-231 cells showed increased expression of ATG5; however, the expression of ATG3 and beclin1 was only slightly reduced at 24 and 48 h post treatment. Expression of the autophagy-related protein Rb and its phosphorylated form p-Rb780 was further investigated in MCF-7 and MDA-MB-231 cells following E2 treatment. As shown in Figure 3B,C, the expression of Rb was significantly increased in MCF-7 cells treated with 20 nM E2 at 6 and 12 h and then significantly decreased afterwards. In contrast, the expression of pRb780 was at background levels at 6 and 12 h/20 nM E2 but then significantly decreased. Observed changes in pRb780 expression was less evident in MDA-MB-231 as compared with that in MCF-7 cells. In that, the expression of Rb protein was slightly reduced at 20 nM E2/6 and 48 h. Moreover, the expression of pRb780 was significantly reduced at 6, 24, and 48 h.

E2 induces mitochondrial membrane depolarization and damage: E2 treatment resulted in a significant mitochondrial membrane depolarization in MCF-7 and MDA-MB-231 cells especially at 20 nM E2/48 h (Figure 4A). MCF-7 and MDA-MB-231 cells treated with 20 nM E2 also showed increased expression of the autophagy-related proteins Pink1 and LAMP2, increased LC3-I/II conversion rate and reduced expression of Beclin1 as compared with control cells or with cells treated with 20 nM of the autophagy inhibitor bafilomycin (Figure 4B). Bafilomycin treatment increased the expression of Beclin1 in MCF-7 cells but not in MDA-MB-231 cells; it also did not affect the expression of LAMP2 in either cell line. Cells co-treated with E2+bafilomycin showed a significant block of the effects of E2 on Pink1 and Beclin1 but not on LC3-I/II expression. E2+bafilomycin co-treatment also resulted in higher levels of expression of LAMP2 as compared with controls or with bafilomycin alone treated cells, especially in MDA-MB-231 cells.

To further evaluate the capacity of E2 to induce mitochondrial damage and autophagy, the expression of markers of mitochondrial damage were evaluated in mitochondrial protein lysates isolated from MCF-7 and MDA-MB-231 cells following E2 treatment. As shown in Figure 4C, E2 treatment resulted in a slight increase in LAMP2 expression in MCF-7 cells at 24 h post treatment and a significant increase in its expression in MDA-MB-231 cells at 48 h post treatment. The expression of Pink1 increased in MDA-MB-231 cells at 24 h post treatment and in MCF-7 cells at 48 h post treatment. Upregulated expression of NOX4 was evident in MCF-7 cells at both time points and in MDA-MB-231 cells at 48 h post treatment (Figure 4C).

E2 treatment induces lysosomal accumulation and damage: Appearance of senescence-associated heterochromatin foci (SAHFs) and increased presence of cytoplasmic LC3-I/II^+^ auto-phagosomes in E2-treated cells (Figure 1A) suggested cell senescence. To further investigate this possibility, E2-treated cells were stained with the lysosomal tracker and analyzed for lysosomal accumulation. As shown in Figure 5A,B, E2 treatment resulted in a significant increase in lysosomal accumulation in MCF-7 and MDA-MB-231 cells at 24 and 48 h post-treatment. Furthermore, increased activity of β-galactosidase, an important biomarker of replicative senescence was evident in cells at 20 nM E2/24 and 48 h post-treatment (Figure 5C,D). MCF-7 cells exhibited much higher levels of SA-β-Gal activity than MDA-MB-231 cells following treatment with 20 nM E2. The ability of E2-treated cells to express pro-inflammatory cytokines was also assessed as an additional marker of cell senescence. As shown in Figure 5E, although the expression of IL-6 significantly decreased in E2-treated MCF-7 and MDA-MB-231 cells at 24 h, it significantly increased in both cell types at 48 h post treatment. The expression of TNF-α did not significantly change in either cell type at either time point.

## 4. Discussion

Previous work has reported on the anti-cancer effects of E2 in endocrine-related cancers [5,6,7,8,9,10,11,12]. Recently, E2 was reported to negatively impact cancer cell growth and metabolism by disrupting cellular iron metabolism, which induces oxidative stress, cell cycle arrest and cellular damage but not apoptosis [15,16,25]. In line with these observations, data presented here suggest that E2 negatively impacts breast cancer cell growth and metabolism through its ability to induce mitochondrial dysfunction-associated autophagy and senescence. This is based on the observation that E2 disrupted mitochondrial function, induced mitochondrial and lysosomal accumulation and flattening, increased SA-β-Gal activity, upregulated IL-6 production and differentially altered the expression of key cell cycle- and autophagy-related proteins. The ability of E2 to induce autophagy and senescence in cancer cells is further evidence of its anti-cancer potential and is in line with a significant body of evidence that has documented the association between anti-cancer chemotherapy and radiotherapy and the induction of autophagy [26,27,28,29,30] and/or premature senescence [31]. It is also consistent with the observation that cancer cells with defective apoptotic potential often respond to chemotherapy by entering cell cycle arrest and senescence [15,27].

E2, whether synthesized within the mitochondria, diffused across the lipid bilayer [32] or recruited as an E2/estrogen receptor-alpha (ER-α) or ER-beta (ER-β) complex [33,34], affects mitochondrial function and structure [35]. E2 has been reported to inhibit mitochondrial inner membrane ion channel activities and respiration [24]. E2 has also been shown to modulate mitochondrial ROS formation [36] and alter mitochondrial membrane potential (ΔΨm) by increasing Ca^2+^ concentration [37]. Treatment of rat liver cells with high dose E2 (>30µM) was reported to inhibit Ca^2+^-induced permeability transition in the mitochondria [24]. While ER-α was shown to associate with E2-mediated increase in mitochondrial respiratory chain (MRC) and antioxidant proteins [38,39], ER-β was reported to reduce mRNA expression of nuclear-encoded subunits of the MRC complex in the vasculature [38]. Moreover, E2 signaling was reported to inhibit radiation-induced cytochrome C release, decrease mitochondrial membrane potential and apoptotic cell death in MCF-7 cells [40]. Overall, these observations are consistent with the finding that E2 treatment resulted in mitochondrial membrane depolarization (Figure 4A).

It is worth noting that previous studies have suggested that E2 signaling associates with mitochondrial survival and function [15,16,26]. E2 was reported to inhibit cell senescence [41] in endothelial progenitor cells [42], mammary epithelial cells [20], mesenchymal stem cells, chondrocytes [43], and brain cells [44]. However, there is evidence to suggest that E2 also induces senescence in certain cell types. For example, significant levels of cell senescence develop during the progression of E2-induced pituitary tumors [19]. Several phytoestrogens and phytoestrogen-like molecules (curcumin, genistein, and quercetin among others) have been shown to induce cell senescence in cancer cells [45,46] through the nuclear factor erythroid-derived 2 related factor 2 (Nrf2) signaling [47,48,49]. E2 itself also was reported to induce anti-oncogenic effects in MCF-7 cells by increasing the activity of Nrf2 through the activation of the PI3K pathway [50].

Our data show that the expression of several autophagy-related proteins (ATG3 and beclin) is differentially altered in E2-treated MCF-7 cells (Figure 3A). Previous work has established that cell senescence associates with increased expression of autophagy-related genes *ATG2B*, *LC3B,* and *GABARAPL2* and *Ulk3* [51,52,53]. Upregulated expression of the *TP53* gene, which promotes autophagy by acting on targets is consistent with previous work which have reported that engagement of ER-β with E2 or ER agonists increases *TP53* gene expression [54]. Additionally, while overexpression of *ULK3* was reported to induce autophagy and senescence, inhibition of *ATG5/7* was shown to delay cell senescence [55]. This is further supported by the fact that multiple autophagy related genes including the MAP1LC3-II are upregulated in E2-treated MCF-7 cells; a finding that is consistent with several previous studies [reviewed in 18]. Moreover, it has been recently suggested that induction of autophagy in cancer cells could itself re- or de-regulate the cell cycle by modulating the expression of cell cycle-related genes as means of cell survival and resistance to chemotherapy and/or preventing apoptosis [56]. Hence, the exact implications of increased expression of cell cycle genes following E2 treatment and whether this could be considered as a trigger or a consequence of autophagy remain ambiguous.

The general consensus is that the expression and activation of P53 and Rb is critical for the induction of senescence [57,58]. Our data show that the expression of P53 and Rb expression increased, mainly in MCF-7 cells early on but then decreased afterwards. Moreover, the data show that pRb was expressed at the background level when cells were treated with 20 nM E2 at 6 and 12 h, but then significantly decreased (Figure 3B). At the risk of over interpreting this set of data, it is possible that such time-dependent changes may reflective a dynamic and complex system that involves opposing players which employ gene suppression, protein degradation and/or activation/inactivation events [18,57]. Additionally, several studies have shown that while P53 is essential for the initiation of autophagy [57,58], its expression and/or structural integrity are compromised by the action of ER-α [54]. Additionally, several studies have suggested that loss or inactivation of Rb [58,59,60,61] and P53 [59], as tumor suppressors, induces degenerative autophagic responses. This may help explain the finding that both proteins were downregulated in E2-treated MCF-7 and MDA-MB-231 cells. Furthermore, inactivation of P53 or pRb often associates with cell-cycle re-entry [61,62] and/or re-initiation of DNA synthesis [63,64]. Therefore, it is possible that E2-induced senescence, as reported here, is reversible [60,62,63] and that suppression of P53 by autophagy may promote tumor progression and prevents tissue degeneration [[60] and references therein]. This is more likely to apply to MDA-MB-231 than MCF-7 cells given that the expression of P21, a critical P53-induced regulator of senescence and inhibitor of cell cycle [64], upregulates early and transiently in MDA-MB-231 cells. Consistent with this observation is the fact that MDA-MB-231 cells have significant proliferative and metastatic potential [65], poor responsiveness to anti-growth chemotherapy [66] and reduced susceptibility to autophagy [67]. Moreover, reversal of senescence occurs more often in cells like MDA-MB-231 as they are incapable of repressing rRNA transcription [68] due to reduced expression of the JmjC domain-containing histone demethylase 1B (JHDM1B) [69].

The idea that E2-induced autophagy and senescence precipitates anti-cancer effects is inconsistent with the observation that E2-induced autophagy enhances tumor progression [47] and that induction of premature (reversible) senescence generates aggressive variants [27]. Other studies have suggested that E2 attenuates autophagy and apoptosis following prolonged hypoxia by Hif-1α-dependent blockage of BNIP3 and IGFBP-3 signaling [70] and that ectopic expression of ERα reverses senescence-like phenotypes (reduced pRb and increased activity of SA-β-Gal) in immortalized human mammary epithelial cells [20]. Our idea is also in disagreement with the observation that autophagy leads to tumor progression [26,27,28,29], especially when the cell cycle is irreversibly blocked [26,71] and the observation that senescent cells often upregulate the expression of tissue remodeling mediators (pro-inflammatory cytokines, growth factors, and matrix metalloproteases) [72].

## 5. Conclusions

Data presented here suggest that E2 signaling disrupts mitochondrial function and induces cellular senescence and autophagy in breast cancer cells. Further work is still needed to investigate whether E2 signaling is capable of precipitating similar effects in other forms of human cancer and whether such effects can be replicated in vivo.

## Figures and Tables

**Figure 1 biology-09-00068-f001:**
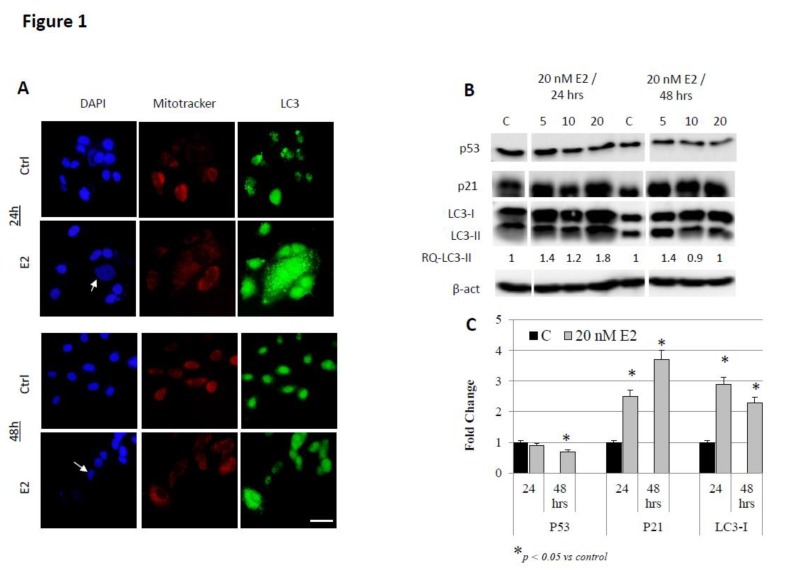
Mitochondrial accumulation and autophagy in E2-treated MCF-7 cells. (**A**) Cells treated with 20 nM E2 or left untreated for 24 or 48 h were stained for DNA (DAPI; blue), mitochondrial accumulation (mitotracker; red) and LC3 (green); images were taken at 40X magnification; scale bar (white) represents 10 μm. Presence of SAHF (senescence-associated heterochromatin foci) in the nuclear region of senescent cells is indicated by white arrows. (**B**) Expression of cell senescence and autophagy markers P53, P21, and LC3-I/II in lysates of cells treated with 20 nM E2 for 24 and 48 h; RQ refer to LC3-II quantity under different treatment conditions relative to control. (**C**) Mean + SD fold change in protein expression in treated and untreated cells based on three separate experiments.

**Figure 2 biology-09-00068-f002:**
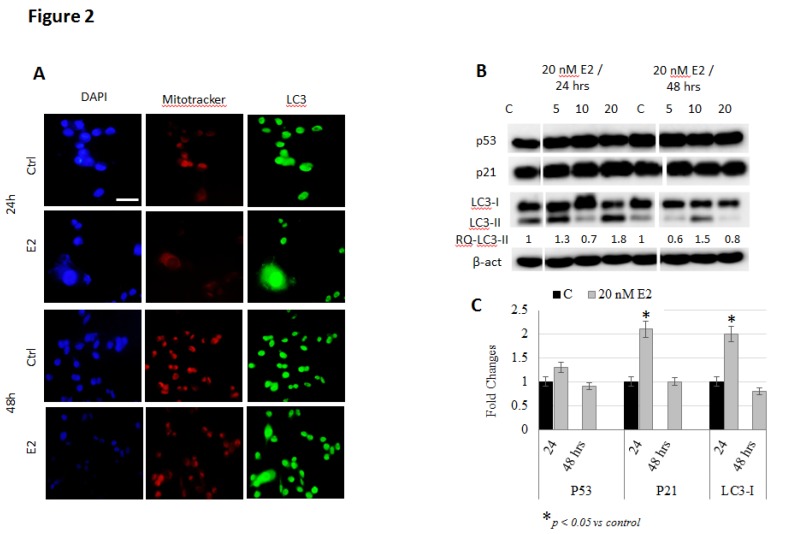
Mitochondrial accumulation and autophagy in E2-treated MDA-MB-231 cells. (**A**) Cells treated with 20 nM E2 or left untreated for 24 or 48 h were stained for DNA (DAPI; blue), mitochondrial accumulation (mitotracker; red) and LC3 (green); images were taken at 40X magnification; scale bar (white) represents 10 μm. Presence of SAHF (senescence-associated heterochromatin foci) in the nuclear region of senescent cells is indicated by white arrows. (**B**) Expression of cell senescence and autophagy markers P53, P21 and LC3-I/II in lysates of cells treated with 20 nM E2 for 24 and 48 h; RQ refer to LC3-II quantity under different treatment conditions relative to control. (**C**) Mean + SD fold change in protein expression in treated and untreated cells based on three separate experiments.

**Figure 3 biology-09-00068-f003:**
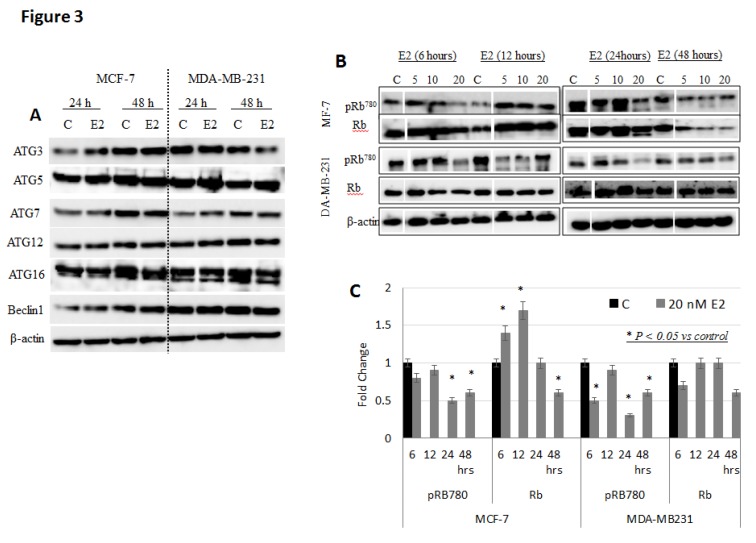
Expression pattern of autophagy-, cell cycle- and senescence-associated proteins in E2-treated MCF-7 cells. (**A**) Expression of autophagy related proteins ATG3, ATG5, ATG7, ATG12, ATG16, and Beclin1 in MCF-7 and MDA-MB-231 cells treated with 20 nM E2 for 24 and 48 hr. Data shown is representative of two separate experiments. (**B**) Expression of Rb and its phosphorylated form p-Rb780 was assessed by Western blotting in lysates of MCF-7 and MDA-MB-231 cells treated with 20 nM E2 for 6, 12, 24 or 48 h. (**C**) Mean + SD fold change in Rb and p-Rb780 expression levels in treated and untreated cells based on three separate experiments.

**Figure 4 biology-09-00068-f004:**
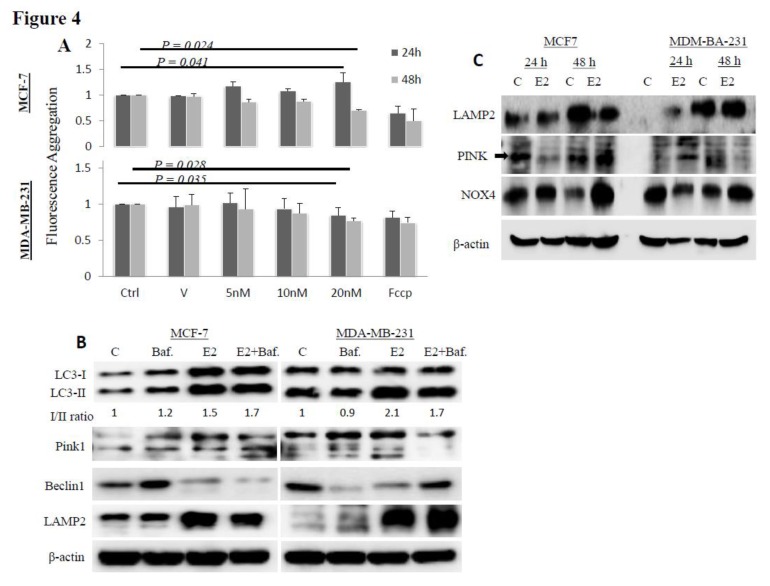
Mitochondrial membrane potential (Δψm) in E2-treated cells. (**A**) Mean + SD of MMP in MCF-7 and MDA-MB-231 cells treated with 20 nM E2 for 24 and 48 h. FCCP-treated cells served as a positive control and untreated cells served as a negative control. Data shown is based on three separate experiments/cell type. (**B**) Expression of the autophagy proteins LC3-I/II and Beclin1, the mitochondrial integrity regulator Pink1, and the lysosome-associated membrane protein 2 (LAMP2) in MCF-7 and MDA-MB-231 cells treated with 20 nM bafilomycin, 20 nM E2 or both for 24 hr. (**C**) Expression of LAMP2, Pink1 and NADPH oxidase 4 (NOX4) in mitochondrial protein lysates isolated from MCF-7 and MDA-MB-231 cells at 24 and 48 h post treatment with E2. Data shown in B and C is representative of three separate experiments each.

**Figure 5 biology-09-00068-f005:**
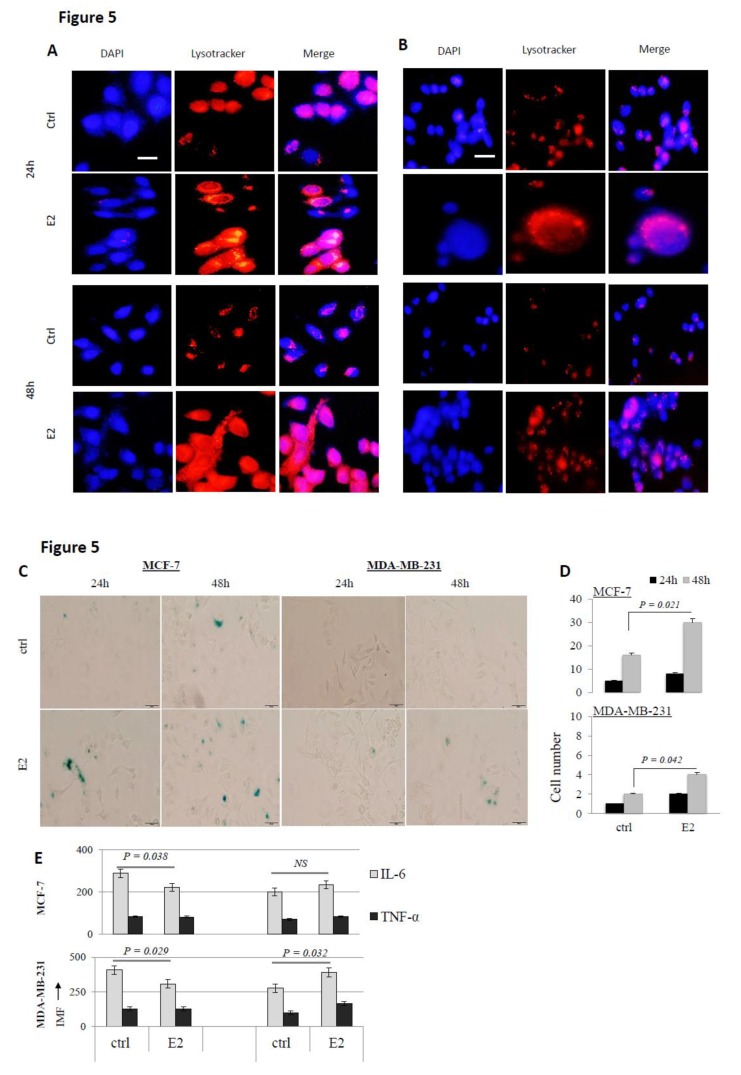
Functional assessment of autophagy in E2-treated cells. MCF-7 (**A**) and MDA-MB-231 (**B**) cells treated with 20 nM E2 or left untreated for 24 or 48 h were stained for DNA (DAPI; blue) or lysosomal accumulation (lysotracker; red). Data shown is representative of three independent experiments. (**C**) Qualitative assessment of senescence in MCF-7 and MDA-MB-231 cells treated with 20 nM E2 for 24 and 48 was done by observing the formation of green-metallic color following the addition of SA-β-Gal reagent. (**D**) Mean + SD of manually counted senescent MCF-7 and MDA-MB-231 cells at 24 and 48 h post E2 treatment in three separate experiments. (**E**) Cells treated with E2 (20 nM) or left untreated were assayed for the expression of cytoplasmic IL-6 and TNF-1α at 24 and 48 hr post E2 treatment. Data shown represent the average + SD of MFI based on three separate experiments.

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
