# Peer review of "Estrogen Signaling Induces Mitochondrial Dysfunction-Associated Autophagy and Senescence in Breast Cancer Cells"

_biology, 2020, doi:10.3390/biology9040068_

Round 1
Reviewer 1 Report
The authors wrote an interesting paper in which they proved that estrogen caused a mitochondrial damage that led to senescence and autophagy in MCF-7 and MDA-MB-231 human breast cancer cells. The rationale was provided and the structure is well-organized. The introduction provides sufficient background, the research design is reasonable, the methods used are suitable. Some of the conclusions are controversial and are in disagreement with the already published data. However it would be necessary to write a review to explain these discrepancies. The paper would be more valued if published some years earlier, the authors cited 73 references, 12 published before 2005, 36 published in 2005-2015, and 24 coming from the recent 5 years. Nevertheless, the subject has not been fully elucidated jet.
The authors should more carefully use the word “adverse” in context of susceptibility of cancer cells to chemotherapeutic agents (e.g., in lines 19-20, either delete the word “adverse” in the line 19 or “such” in the line 20). Can the events leading to cancer cell death be considered as causing the adverse effects? Adverse has a negative meaning, harmful, unfavorable.
In all the experiments performed E2 was used in the concentration of 20 nM only, while in the lines 69 – 71 is written “Cells were then treated with increasing concentrations of E2 and …” what suggests that a series of concentrations were applied. The choice 20 nM of E2 should be explained. Was the toxicity (MTT, SRB, LDH) of E2 at 20 nM determined? Was it identical for both cell lines?
In the microarray experiment GeneChip Rat Gene 2.0 ST was used, a chip which is designed to study gene expression patterns in rats. Of course rats are often used as model organisms for studying various processes in humans, but in the case when cultured human cells are investigated the human gene chips should be applied. Otherwise the obtained results can be doubtful.
The microarray data were shown for the MCF-7 cell line only. What about the MDA-MB-231? It has been widely documented that both cell lines have different nature and susceptibility to anticancer drugs.
It is advisable to place an additional column in the Table 1 showing the fold ratio of the differentially expressed genes vs. control and the statistical significance. Having in mind these values we can decide whether the results should be verified by e.g. qRT-PCR or correlated with the encoded protein expression.
The microarray analysis seems to be the weakest part of the study and without loss of its significance can be deleted.
Minor mistakes:
The gene names and symbols style must be consistent in the whole manuscript and adhere to the commonly accepted guidelines. Do not use superscripts, italicize gene symbols, CAPITALIZE for human and non-human primates, etc.
Line 81 - The word “with” is missed, should be written: treated with ethanol.
Moreover, the concentration of ethanol in the culture media should be indicated (I hope that was very low and did not influence the cellular processes investigated).
Line 82 - delete “was used”. Capitalize Aldrich
Lines 159, 290, 298, refs. 27, 25, 26: the references should be cited in the order appearance.
Fig. 2C - the Y-axis title should be corrected: fold changes, not “fold cjanges”.
Line 192 - associated, not “associating”.
Lines 203, 205, 232, 233 - there should be 231 instead 23 in the cell line name.
Line 303- the abbreviations ERα and ERβ ought to be explained, provide full names.
Line 337 - Is there a mistake in the word “upregulates”?? The sentence is difficult to understand. Did you mean that genes, including the MAP1LC3-II, are upregulated in E2-treated MCF-7…..? If so, please correct in the text.
Lines 344/345 - and line 377- cell cycle genes, not”cell cycling”.
Line 375 - the abbreviation for SA-β-Gal is used in this place while the full name of this marker appears several times earlier in the text. The full name should be used only once.
Line 538 - the year of the publication is missed (ref. 56).
Author Response
Please see attached document "Cover letter-Response to Reviewers' comments"

Reviewer 2 Report
Dear Authors,
Thank you very much for the opportunity of reviewing your paper. The work performed and the results you have obtained are very interesting.
I find this article to be of great value.
Cloud you please explain in the manuscript the following aspects:
- Why you decided to use exactly 20nM of E2 ?
- Please indicate on each image which is the mean and which is the SD (for a better understanding of the images) (example: Figure 1 (C))
- Study limitations and further directions
Author Response
Please see the attached document "Cover letter-Response to Reviewers' comments"
